# In Vitro Infant Digestion of Whey Proteins Isolate–Lactose

**DOI:** 10.3390/foods12030667

**Published:** 2023-02-03

**Authors:** Sarizan Sabari, Norliza Julmohammad, Haque Akanda Md Jahurul, Patricia Matanjun, Noorakmar Ab. Wahab

**Affiliations:** 1Faculty of Food Science and Nutrition, Universiti Malaysia Sabah, Kota Kinabalu 88400, Sabah, Malaysia; 2Department of Agriculture, School of Agriculture, University of Arkansas, 1200 North University Dr., M/S 4913, Pine Bluff, AR 71601, USA

**Keywords:** in vitro protein digestion, Maillard reaction, WPI conjugates, lactose

## Abstract

The model in vitro protein digestion technique has received greater attention due to providing significant advantages compared to in vivo experiments. This research employed an in vitro infant digestive static model to examine the protein digestibility of whey proteins isolate–lactose (WPI–Lac). The polyacrylamide gel electrophoresis (PAGE) pattern for alpha-lactalbumin of WPI at 60 min showed no detectable bands, while the alpha-lactalbumin of the WPI–Lac was completely digested after 5 min of gastric digestion. The beta-lactoglobulin of the WPI–Lac was found to be similar to the beta-lactoglobulin of the WPI, being insignificant at pH 3.0. The alpha-lactalbumin of the WPI decreased after 100 min of duodenal digestion at pH 6.5, and the WPI–Lac was completely digested after 60 min. The peptides were identified as ~2 kilodalton (kDa) in conjugated protein, which indicated that the level of degradation of the protein was high, due to the hydrolysis progress. The conjugated protein increased the responsiveness to digestive proteolysis, potentially leading to the release of immunogenic protein by lactose, and to the creation of hypoallergenic protein.

## 1. Introduction

In newborns, food protein digestion is complicated and crucial [1]. An infant has a relatively short digestive tract, and lacks gut microbial mass. It is normally not possible for an infant to absorb sufficient dietary protein. The digestive tract of the infant is sterile, and lacks enzymes to digest proteins and other nutrients. Impairment of protein digestion may lead to poor weight gain, food allergies, and malnutrition [2]. An inadequate protein intake during the first year of life has been linked to impaired growth in later infancy and early childhood. The late introduction of protein into the diet of infants may be a key factor in explaining the higher prevalence of stunting and underweight among infants born in developing countries.

The effect of milk processing on the alteration of base protein has an impact on its digestibility. Protein digestibility has been studied for the past three decades, as a result of public and commercial interest in the importance of protein digestibility in humans [3]. The functional properties of proteins can be improved using non-chemical reactions: this modified protein can be of huge interest to the food industry. The effect of heat treatment on protein digestibility was described by [4]. Generally, the Maillard reaction (MR) is used as a non-chemical reaction that belongs to the first stage of the MR [5]. The chemistry that drives the Maillard reaction is complicated; thus, the sophistication and variety of Maillard reaction products have attracted the attention of researchers in numerous disciplines of study over the years. As a reducing sugar, lactose can participate in the MR. Conjugation of the lactose to the whey proteins isolate requires heating in a certain condition, but does not require chemical usage in reaction in order to accelerate the spontaneous reaction. In the MR, a variety of parameters are implemented, including the temperature applied to the reaction, the type of carbohydrate employed, the pH, and water activity—with temperature and pH playing a critical role in the formation of MR products, as mentioned by [6]; therefore, an optimized condition of the MR can be a notable method for improving the functional properties of the native protein.

Previously, [7,8] investigated the conjugation of whey protein and lactose. They agreed that with the changes in amino acid composition, reducing saccharide sugar with a smaller molecular weight was glycated in larger numbers on each protein molecule [7]; they believed that the isoelectric point of the glycated protein was lowered, and increased the denaturation temperature of the whey protein; therefore, they suggested that the dominant mechanism which controlled the whey protein aggregation and the transparency of dispersions was repulsive steric interactions; meanwhile, they concluded that the protein–lactose ratio initially impacted the reaction when present in whey protein concentrate: whey protein concentrate 35% (with 35% protein and 51% lactose) compared to whey protein concentrate 60% (with 60% protein and 23% lactose) [8].

The MR has advantageous consequences in foods, but its consequences in milk products are negative: for example, the MR can impact the milk protein with brown coloring, off-flavors, loss of nutritive value (lysine), and loss of solubility in milk powder [9], whereas the casein–lactose conjugate has been found to have potential antioxidants for food application [10]. Another study found no clear trace of MR products in the digestive tract of the human body [11]. According to a past researcher in 2007, with high heat and alkaline pH, protein undergoes several chemical alterations, which involve the lysine residue present in protein, consequently reducing the digestibility of the protein [12]. In addition, the reaction of reducing sugars with ε-amino groups can impact on the digestibility of lysine, but thermal denaturation can sometimes render it harmless.

The effects of the modification of protein can alter the primary structure of the protein, leading to unfolding protein, and therefore improving the biological value of the protein: the digestive enzymes can easily access the protein, and this affects the absorption of the protein into the body. However, it has been reported that modification of the primary structure of a protein may lower its digestibility, which may produce residue that is biologically available [13]. This is supported by [12], who revealed that modification of protein can cause unfolding, aggregation or lactosylation: of beta-lactoglobulin in milk: these reactions will affect the digestion and the absorption of the protein, and will also impact the access of the protein into the gastrointestinal immune system in the body.

Therefore, future research on the quantitative analysis of protein modification and digestion in the in vitro infant gastrointestinal tract is needed, to enable a clear process that might help in the development of an infant formula that will have minimum impact on the allergenicity of the protein milk, and improve the digestibility of protein in infants. In addition, it is necessary to investigate the digestibility of conjugated whey proteins with lactose, to achieve the creation of hypoallergenic food protein and the masking of immunogenic peptides by lactose. The objective of this study was to understand the digestibility of conjugated whey proteins isolate–lactose in a simulated infant gastrointestinal tract, using a static protocol in vitro infant digestion model with the optimized conditions proposed.

## 2. Materials and Methods

### 2.1. Materials

The whey protein isolate (WPI) used in this study was purchased from Nutrija Lifesciences in India. In order to ensure that the protein quality met international standards, whey isolate from the Davisco Food (BiPRO, Eden Prairie, MN, USA) was used in the production of Nutrija’s Whey Protein Isolate. WPI (BiPRO) was manufactured by cold processing and micro-filtering. The physical appearance of this product was a cream-color powder. The BiPRO was not denatured, and was soluble over the pH range of 2.0 to 9.0. According to the manufacturer, this WPI comprised 90.0% protein, 0% fat, 1.25% ash, 2.9% moisture, and 0% sugar.

The lactose (Ph. Eur, Sigma-Aldrich/US, Burlington, MA, USA) was finely crystallized white powder. The pH range of this product was 4.0 to 6.5, with 0.5% moisture, and 161 g/L solubility. Lactose can take part in the Maillard reaction as reducing sugar. Alterations in the content of amino acids cause a greater number of the reducing saccharide sugar molecules to be glycated on each protein molecule [7]. The majority of individuals with cow milk allergies are under the age of three: hence, an infant digestive model was used. The method was modified from [14]. Pancreatin from porcine pancreas (Sigma-Aldrich/US, −20 °C, 8 × USP specification) was used instead of trypsin and chymotrypsin, and the duodenal phase was reduced to 100 min. Pepsin from porcine gastric mucosa (CAS 9001-75-6, Sigma-Aldrich/US, 2–8 °C) and pancreatin from porcine pancreas (Sigma-Aldrich/US, −20 °C, 8 × USP specification) were purchased from Weitengen Sdn Bhd, MAS. To confirm that any alterations related to the glycates were caused by pepsin and pancreatin, control digestion without pepsin and pancreatin was carried out. All other chemicals and reagents used were analytical grade. Preparation of all solutions employed deionized water, purified by treatment with a Milli-Q (Millipore Corp., Bedford, MA, USA).

### 2.2. Methods

#### 2.2.1. Whey Proteins Isolate–Lactose Conjugation

A covalent bond is formed when an amino group of a protein reacts with a carbonyl group of carbohydrates: this reaction results in the formation of a new modified protein, most notably protein conjugates. Lactose was used in the conjugation with the WPI because it is a reducing disaccharide sugar with a lower molecular weight that reacts as rapidly as a monosaccharide sugar. Protein conjugation with lactose was carried out according to the method [15], and 40 mL of purified water was used to dissolve native WPI and WPI–Lac at 1 to 0.4 (*wt*/*wt*) protein to sugar. The mixtures then were frozen for twenty-four hours and lyophilized to produce a dry powder (Labconco, 4.5 L, 50 °C, Kansas City, MO, USA). The powder was then dried at 40 °C in a desiccator containing a 0.8 water-activity potassium bromide (KBr) solution [15]. The temperature was employed below the denaturation temperature of protein and suitable relative humidity. After incubation, samples were collected for subsequent analysis. This research followed the dry method, because it had the advantage of easily controlling the reaction conditions, free of poisonous and harmful products [5].

#### 2.2.2. Preparation of Glycates Solution In Vitro Digestion

Glycates solution was prepared in 4 mg/mL. Around 0.04 g of WPI–Lac conjugates were dissolved in 10 mL of Milli-Q water, and the mixture was immediately stored at −20 °C.

#### 2.2.3. Simulated Gastric Fluid (SGF)

The SGF solution was prepared using the procedure described in [14]. Approximately 100 mL of 0.15 M SGF was produced using sodium chloride (NaCl) for 10 samples, and the pH was adjusted to 3.0 with 1.0 M HCl. In 100 mL of deionized water, 0.8768 g of NaCl was dissolved and subsequently kept at −20 °C.

#### 2.2.4. In Vitro Infant Gastric Digestion

The digestion mechanism in infants aged 0 to 6 months is in the stomach and intestine phase, due to the very short transit time through the oral passage for infants [16,17]. In vitro infant gastric digestion was carried out in accordance with the method [14] for minor modifications. The WPI and glycate solution concentrations and volumes were 4 mg/mL and 1.5 mL, respectively. The sample solution’s pH was raised to 3 with 0.5 M HCl, and its final volume was increased to 1.98 mL, using SGF. Before being transferred in an Eppendorf tube, the sample solution was diluted with 19 μL of 0.625 mg/mL pepsin stock solution, to achieve a final protein concentration of 3 mg/mL. The digestion process was initiated by setting the temperature and speed at 37 °C and 200 rpm, respectively. Before the digestion proceeded, 100 μL of aliquots were pipetted immediately into 20 μL of 0.5 M ammonium bicarbonate (NH_3_CO_3_), as a control. The period for gastric digesting was regulated to 5, 30, and 60 min. Then, 100 μL of aliquots were directly pipetted into 20 μL of 0.5 M ammonium bicarbonate after 5, 30, and 60 min of stomach digestion (NH_3_CO_3_): these tests were carried out in duplicate. The digesta were immediately frozen, and stored at −20 °C for later investigation (SDS–PAGE). Before beginning duodenal digestion, the pepsin-containing gastric reminder was inactivated by raising the pH to 7 in 10 min [1].

#### 2.2.5. Preparation of Phosphatidylcholine-Bile Salt (PC-BS)

A stock solution of egg L-phosphatidylcholine (PC, c = 100 mg/mL chloroform, MW: 768 g/mol, 99% purity, P2772, Sigma-Aldrich) in 100 μL (0.1 mL) was evaporated at 50 °C under vacuum. Later, the solution was redissolved in 333.3 μL of BS, which was made up of equal parts sodium taurocholate (T4009, Sigma-Aldrich) and sodium glycodeoxycholate. BS was purchased from Weitengen Sdn Bhd (G9910, Sigma-Aldrich). Prior to duodenal digestion, a 40 μL aliquot was added to the reaction, resulting in a final BS concentration of 1.85 mmol/L and a PC concentration of 0.91 mmol/L. The re-suspended PC solution was put into an incubator, and was shaken at 37 °C and shielded from light with a cushion of nitrogen. A PC–BS mix was freshly produced for each digestion [14].

#### 2.2.6. Preparation of Bis-Tris HCL

Bis-Tris HCL was made by dissolving 6.057 g of Tris-Base in 90 mL of deionized water, to give a concentration of 0.5 M. Using 37% HCL, the pH of the solution was adjusted to 6.5. Deionized water was added, to bring the final volume of the solution up to 100 mL.

#### 2.2.7. In Vitro Infant Duodenal Digestion

In vitro duodenal digestion was carried out by the following method [14], with minor modifications. The remainder of the stomach was supplemented with 88 μL of 0.5 M Bis-Tris HCl in pH 6.5 and 40 μL of PC–BS solution. The pH levels were adjusted to 6.5. To begin duodenal digestion, the reaction mixtures first had to be warmed to 37 °C, and 32 μL of pancreatin at a concentration of 5.899 mg/mL added, before being placed in a shaking incubator at 37 °C with 200 rpm. After 5, 30, 60 and 100 min, 200 μL of samples were collected. Enzyme activity was stopped by pipetting aliquot vials containing 0.5 μL of 2-mercaptoethanol and 25 μL of Laemmli sample buffer. The aliquots were then heated to 100 °C for 3 min. The aliquots were cooled on ice, and kept at −20 °C as soon as the mixture in the vials reached 100 °C, which took 40 s [14].

#### 2.2.8. Sodium Dodecyl Sulfate–Polyacrylamide Gel Electrophoresis (SDS–PAGE)

Three mini-Protean 3 cells from Bio-Rad were used to perform SDS–PAGE digesta under reducing conditions (Tris-HCl gel, 15% reducing SDS–PAGE). Then, 45 min of electrophoresis was performed at 200 V and 22 °C [1]. In this investigation, Precision Plus Protein^TM^ Dual Xtra Prestained Protein Standards (2–250) kDa were used as a protein marker, and gels were stained with Coomassie blue G-250 (161-0786, Bio-Rad, Hercules, CA, USA) following the procedures of [14], with slight modification.

## 3. Results

A static in vitro infant digestion model was applied, to compare the digestibility of the WPI–Lac conjugates versus the native WPI. In infants, milk protein is the sole source of nutrition for growth and development [18]. In infancy, the digestion of food proteins is a challenging and essential process [19]; the digestibility of protein in the gastrointestinal tract is significantly lower in infants [20]. An infant static digestion model was chosen because one of the most typical and rampant allergens that affect atopic children early in life is milk [21]. Due to the existence of many recognized allergens in cow’s milk, several infants are reluctant to drink it, albeit cow-milk-based infant formula has a long history of infant feeding in infant nutrition. Cow’s milk allergy (CMA) has been recognized as a key contributing factor of allergenicity in a certain population of babies. The method of in vitro infant digestion was modified from [14], with minor alterations. The gastric digestion was set to 60 min, and the duodenal digestion was reduced to 100 min. Next, a control digestion in the gastric and duodenal phase was carried out without the presence of pepsin and pancreatin, as shown in Figure 1, Figure 2, Figure 3 and Figure 4. Without the pepsin and pancreatin, the WPI and WPI–Lac conjugates are shown unaltered at the end of gastric and duodenal digestion.

### 3.1. Gastric Digestion

Gastric digestion primarily degraded the native WPI α-Lac, and the intensity of the bands at a molecular weight of approximately 14 kDa (Figure 1) slightly decreased. No detectable bands of α-Lac protein were shown toward the end of 1 hour of gastric digestion at pH 3 (Lane 5). The WPI without pepsin was shown unaltered during gastric digestion, as presented in Lane 1. Meanwhile, the β-Lg shown in the SDS–PAGE picture significantly resisted in the whole 60 min of gastric digestion with or without pepsin.

In Figure 2, the band of α-Lac of conjugated protein, was completely digested in Lane 3 after the first 5 min, and no intact α-Lac protein was detected after 60 min (Lane 5). As observed in Figure 2, high molecular weight species increased band intensity, which indicated that the conjugation of WPI and Lac had occurred. Meanwhile, the β-Lg of the conjugated proteins was similar to the β-Lg of the native WPI. The proteolysis by pepsin during 60 min of gastric digestion in static in vitro infant digestion showed the β-Lg to be insignificant at pH 3.0.

### 3.2. Duodenal Digestion

The PAGE pattern of the native WPI, after 60 min of gastric digestion with pepsin and 100 min of duodenal digestion in the presence of pancreatin or without pancreatin and PC–BS at pH 6.5, is shown in Figure 3. The native WPI contained α-Lac and β-Lg with molecular weights of approximately 14 kDa and 18 kDa. By increasing the digestion time in the intestines, the intensity of the band for α-Lac decreased, and was completely undetectable after 5 min of digestion, as shown in Lane 3. Nonetheless, the β-Lg in the native protein was partially digested, which may have been due to limited access to pancreatin cleavage sites on the heat-induced native protein structure, whereas at 100 min of digestion, the β-Lg showed no intact protein (Lane 6).

In the early digestion time, the α-Lac and β-Lg of the conjugated protein partially hydrolyzed indicated the presence of pancreatin and PC–BS (Figure 4). The degradation of conjugated proteins was completely digested after 60 min, as shown in Lane 5 in the duodenal phase. After the duodenal digestion phase, a large amount of a small number of peptides in Figure 4 was identified at approximately 2 kDa (Lane 6) in the conjugated protein, which indicated that the level of degradation of the protein was higher due to the optimized condition applied to the sample, and as the hydrolysis progressed.

## 4. Discussion

### 4.1. Gastric Digestion

An in vitro infant digestion static model was applied, to compare the digestibility of the native WPI versus the conjugated WPI–Lac. Static infant digestion models have been used to answer a variety of scientific questions, including digestibility and bio-accessibility [22]. Gel electrophoresis is a simple method for visualizing the early stages of protein digestion and the production of large peptides [23]. The loss of remaining intact proteins is nonetheless visible on SDS–PAGE: the extent of the disappearance of intact proteins—not the extent of peptide bond hydrolysis—is revealed by SDS–PAGE, but only to the extent of the single cut in the polypeptide chain that is necessary for the intact protein to no longer be visible on SDS–PAGE gels [24]. As there was no whey-proteins-isolate-to-lactose ratio, the time required for the protein to be digested increased during the stomach digestion phase. The native whey proteins were resistant to delayed gastric emptying or gastric coagulation, therefore there appeared to be no reciprocation of the whey-proteins-isolate-to-lactose ratio to the native whey protein coagulation. This finding was in line with the previous finding [14], but contrary to [25], which stated that α-Lac completely resisted proteolysis by pepsin during gastric digestion. Researchers have previously discovered the limited digestion of α-Lac under simulated gastric digestion. Investigation of the in vitro digestibility of α-Lac infant formula in the stomach at pH 1.5–4.0 found that α-Lac was hydrolyzed at pH 1.5–2.5, but was resistant to proteolysis at pH values greater than 3.0 [26]; meanwhile, α-Lac resisted in vitro digestion significantly [27].

In Lane 1 (Figure 1) the WPI did not alter the amino group content at 0 min digestion when SGF was added. This was predicted, because pepsin enzymes were not introduced. At this point, the free amino groups assessed consisted primarily of the amino group of the side chain of lysine, and the N-terminal amino groups of the proteins and peptides [24]. After 60 min of gastric digestion (Figure 1), greater pepsin-induced hydrolysis was detected in comparison to gastric time 0 min samples, which can be linked to no detectable α-lac bands toward the end of one hour of gastric digestion at pH 3. On the other hand, the β-Lg in the SDS–PAGE picture showed significant resistance throughout the 60 min of gastric digestion. Past studies [15,25,28,29] have also pointed out the same result found in this study, whereas [30] noted that β-Lg is stable to denaturation and resistant to proteolysis, like other allergens. The insignificant digestion of β-Lg in gastric digestion is due to the presence of phosphatidylcholine (PC) in the gastrointestinal tract [31]. In addition, the existence of two intramolecular disulfide bridges in the β-Lg also accounts for its greater resistance to digestion [32].

In this study, the native WPI was conjugated with disaccharide sugar instead of polysaccharide sugar. The MR with polysaccharides limited the reaction, and generated a high Schiff base concentration early in the MR pathway [33,34]. Interestingly, reducing sugar with smaller molecular weight is more reactive in the conjugation of protein compared to polysaccharide sugar [35]. As expected, the whey-proteins-isolate-to-lactose ratio decreased the time of protein digestion in the gastric phase. The band of α-Lac of conjugated protein, after the first 5 min in Lane 3, was completely digested, and no intact protein was detected after 60 min (Figure 2). The shorter the carbonic chain of the sugar, the greater the number of open chain forms, and the greater the sugar’s reactivity to the amino groups of proteins [36]; thus, due to heating and carbohydrate binding, the unfolding of globular whey proteins may have rendered concealed cleavage sites more accessible; therefore, for whey proteins, a balance between protein unfolding and steric hindrance, depending on the precise processing conditions, will result in an increase or a decrease in digestibility [37]. The resistance of whey proteins to gastric proteolysis is believed to be related to their conformation, which is necessary to exert certain physiological functions [37]. β-Lg can trigger allergenicity, which mostly occurs in children below two years of age, due to stability in denaturation and resistance in proteolysis, as observed in an earlier study by [30]: thus, β-Lg is mostly undigested during gastric digestion. Many in vitro investigations have focused on β-Lg hydrolysis. Early studies have shown that unfolding and aggregation of whey proteins enhanced the accessibility of protein cleavage sites by heating more than 75 °C: thus, β-Lg was denatured, and became more digestible by pepsin in vitro [38,39,40].

The most common cow’s milk protein allergy has been linked to the abundance of β-Lg in whey proteins (CMA). CMA has been found to be one of the main causes of allergies in a certain group of infants, and has been thoroughly studied [41]. CMA is the most common food allergy in children less than 1 year old, with a prevalence of 1.9–4.9% [42]; however, CMA is most noticeable in infancy, and exhibits a considerable percentage, of 2–3%, between 0 and 3 years [43]. After 15 years of investigation, milk allergy terminology was diagnosed in 2018 [44]: 13% to 76% of this protein is related to the prevalence of allergies [45], and β-Lg is indicated as an allergen [21]. Because of its stability in the acidic gastric environment, and its compact structure, β-Lg—like many other food allergies—resists gastric digestion. This could increase the number of bigger fragments that are exposed to the gut-associated lymphoid tissue, potentially increasing the risk of an allergic reaction [46].

Modified protein can cause the β-Lg to unfold or aggregate: this may lead to an impact on protein digestion and absorption, as well as protein accessibility to the gastrointestinal immune system in the body [47]. This finding is consistent with the findings of past studies by [48], in which there was a direct correlation between the presence of β-Lg and the production of CMA. The gastrointestinal tract also hosts a significant population of immune cells and lymphoid tissue, to aid in digesting, and because it is a significant pathogen entrance point. Therefore, it is crucial to investigate how food proteins, such as β-Lg, interact with the immune system, based on a thorough understanding of its digestibility, which can be altered by the type and degree of heat processing that has been employed: for instance, unfolding might reveal digestive enzyme cleavage sites that were hidden in a protein’s natural form. β-Lg is sensitive to stomach digestion, when heated to about 70 °C in solution [49]. In vivo analysis shows that native β-Lg is resistant in the gastric phase of digestion, but is digested in the intestinal phase of digestion, as mentioned in [50]. In the infant stage, gastric proteolysis is much lower compared to an adult. The optimum pH for pepsin activity ranges from 1.6 to 4, according to [51]. In this study, gastric digestion was set up to pH 3. The digestive tract in an infant is immature, and due to the gastric acidic fluid, protein denatured and low gastric pH will lead to activation of pepsin. High gastric pH will decrease output pepsin: therefore, it inhibits the digestion of protein in infants. A similar finding was also reported in the studies of [3], which showed that protein digestion is affected by gastric pH, and that gastric protease pepsin is only active at low pH. On the basis of the comprehensive literature review, the gastric phase in a normal infant’s secreted gastric juice is less acidic, and ultimately reaches a certain degree of acidity in the infant’s stomach: this is supported by a past study, which revealed that gastric juice secretion showed that gastric enzymes of pepsin play an important role in the hydrolysis of protein [52].

### 4.2. Duodenal Digestion

Following digestion in the gastric phase in the infant’s stomach, the chyme—which includes the milk-based meal and gastric secretion—is progressively emptied into the intestine. An infant’s intestine consists of three parts: the duodenum; jejunum; and ileum. The digestion and absorption of nutrients from the milk-based meal takes place in the intestinal phase. The proteolysis of protein in the intestinal phase has been investigated, due to its contribution to the digestion of protein in humans: the first part of the intestine, which is the duodenum, has a major role in the hydrolysis of protein, as mentioned by [53], who claimed that it was dependent on the pancreatic proteases along with the mucosal brush border and cytosolic peptidase.

The rate of digestibility of conjugated protein in duodenal phase digestion is demonstrated in Figure 4. As observed in this study, the digestibility of modified protein under simulated gastric and intestinal digestion has also been investigated by researchers: to increase food protein function, they investigated whey protein glycates with reducing saccharides through MR [7]. The authors reported that, after glycation, digestion of WPI significantly decreased in the simulated gastric phase, but increased in the intestinal digestion phase: this was due to the resistance of β-Lg in the gastric environment, as reported by [54,55,56]; however, in 2014 studies, no significant digestion of β-Lg was reported [57]. On the other hand, β-Lg was completely digested at the end of intestinal phase digestion [55]: this was explained by the covalent linkage created when carbohydrates were coupled to the free amino groups of amino acids, particularly lysine, during MR [58]. Maillard Reaction Products (MRPs) can improve protein digestion relative to native protein in the digestive tract.

MR can increase or decrease protein digestibility through the modification of protein properties, according to [59], who found out that stearic hindrance formation, caused by the carbohydrate moiety, influenced the digestion of MRPs. Surprisingly, glycated lactoferrin, in their research, showed susceptibility to proteolysis, which was affected by the MR. The conjugation of protein with lactose enhances the susceptibility to digestive proteolysis, by creating more accessibility for the enzymatic digestive attack on the cleavage sites of the protein. The digestibility of protein glycates in the gastrointestinal tract is believed to reduce the allergenicity of protein [5,15,33,60]. When β-Lg passes the intestinal epithelial barrier, it may be exposed to immune cells, which, in the case of heat-treated β-Lg, may cause the immune system to become activated [61]. Depending on the precise heating parameters and processing, the β-Lg’s digestion is increased; however, after end-intestinal digestion, a slightly undigested form of β-Lg still existed [47], which was in line with our observation (Figure 4). At the end of proteolysis in the gastrointestinal tract, the undigested protein still remained. Thus, scholars have, for decades, devoted much attention to attempting to modify protein structure, so as to eliminate the allergenicity of protein. To reduce allergenicity, milk proteins have been glycated with polysaccharides of varying molecular weight, under a variety of circumstances [62]. Hypoallergenic protein development has been explored [14]. β-Lg was modified by conjugation with dextran (polysaccharide sugar). Surprisingly, modified β-Lg is digested in glycates as compared to native WPI, but its digestion is slower in the glycated form. In 2007, a study that examined the implication of using different molecular masses of dextran in the glycosylation of WPs [16] proved that conjugation of β-Lg with dextran 10 and 20 kDa improved its solubility at pH 5 but reduced it at pH 4: this explained the findings of [14] when, throughout the entire gastric phase at pH 3, the glycate of β-Lg remained undigested, but in the duodenal phase at pH 6.5 the SDS-PAGE bands of glycate left a very faint band. WP-dextran 150 kDa (WD150) showed a slightly lower duodenal digestion rate compared to WD10, which may have been the result of increased steric hindrance for WD150; thus, the access of enzymes into the conjugated protein was limited.

Protein digestibility can generally be increased or decreased by thermal processing, depending significantly on the heating conditions used, which dictate the structural protein alterations. The majority of in vitro investigations have found that milk’s protein digestibility increases after thermal processing, possibly as a result of the heat-induced unfolding of the globular whey protein [37]; therefore, the main structure of a protein can be altered by the effects of protein modification. As a result, the protein unfolds, increasing the biological significance of the protein. Protein is easily accessible to digestive enzymes, which affects protein absorption in the body.

## 5. Conclusions

The present study underlined how the impact of protein conjugation, under optimized conditions through MR, increased susceptibility to protein degradation in in vitro infant digestion. α-Lac in WPI-native showed complete resistance to proteolysis by pepsin enzyme in early gastric phase digestion, while conjugated protein with lactose was rapidly hydrolyzed after the first 5 min of digestion, and no intact protein was detected after 60 min of digestion. In duodenal digestion, the hydrolysis of β-Lg resulted in WPI-native and WPI-lactose partially hydrolyzed by pancreatin enzyme, while at 60-to-100 min, the conjugated protein showed no intact protein. Conjugated protein enhances susceptibility to digestive proteolysis, by being more accessible for the enzymatic digestive attack on the cleavage sites of the protein, thus increasing the bio-accessibility and absorption of the proteins. Conjugation may alter the protein structures by reducing the available lysine, providing a “shielding effect” on these conjugated proteins, and creating steric hindrance, which could limit the body’s production of antibodies. Thus, conjugated protein with lactose will possibly create a new generation of whey protein ingredients that could reduce allergenicity in infants. 

## Figures and Tables

**Figure 1 foods-12-00667-f001:**
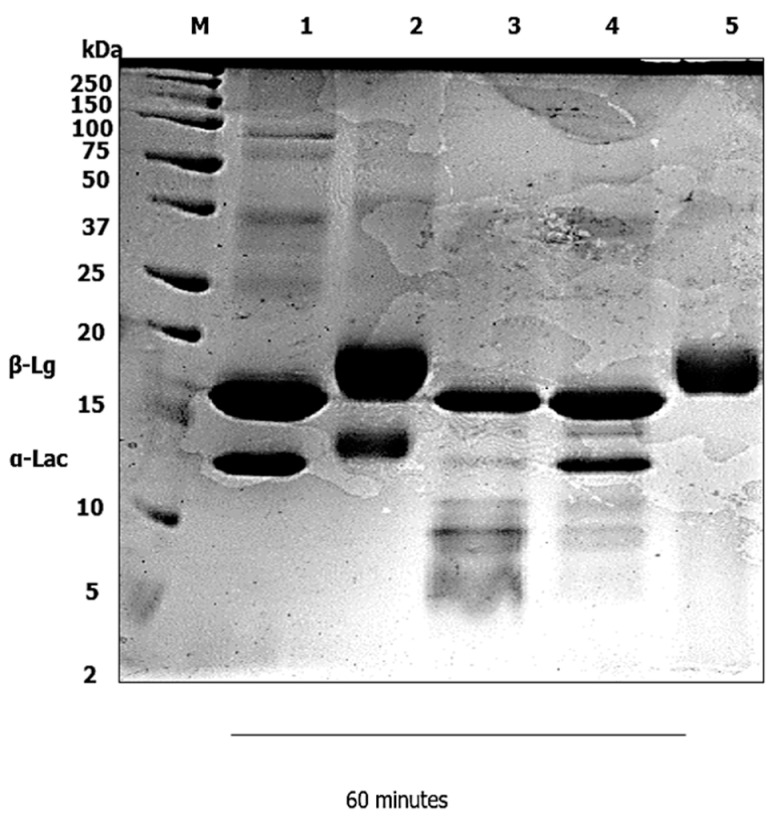
In vitro gastric digestion of native WPI using pepsin. The α-Lac of the native WPI partially hydrolyzed at early 30 min (Lane 4) with the presence of pepsin, and was not detectable at 60 min digestion (Lane 5). The WPI without pepsin was shown unaltered during gastric digestion. The β-Lg was detectable throughout the whole 60 min of gastric digestion. Lane 1: WPI without pepsin; Lane 2: 0 min; Lane 3: 5 min; Lane 4: 30 min; and Lane 5: 60 min. Lane M represents the protein marker.

**Figure 2 foods-12-00667-f002:**
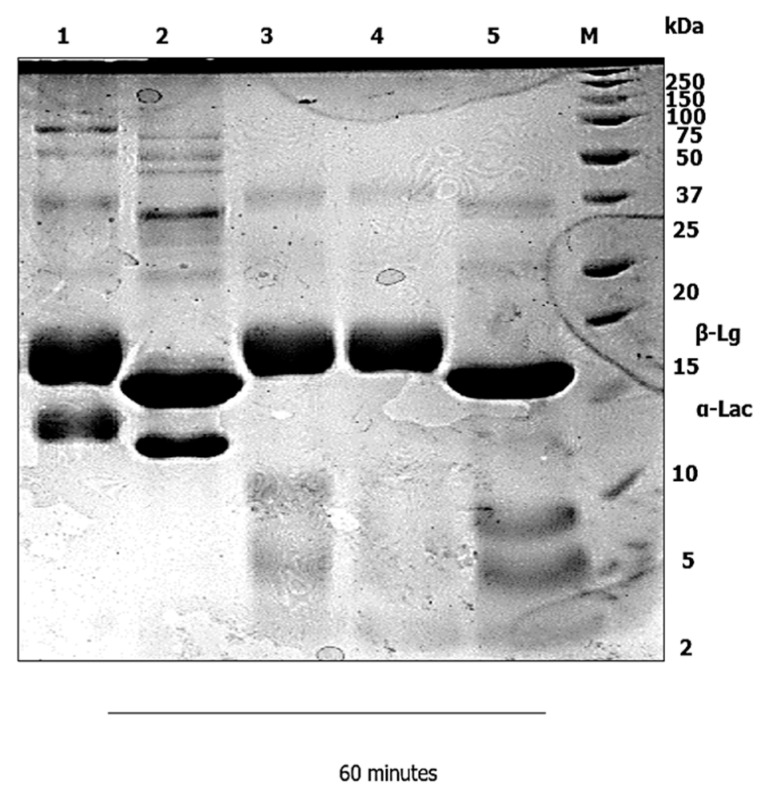
In vitro infant digestion using a static model of WPI-lac. α-Lac in Lane 2 was completely hydrolyzed with pepsin after 5 min of digestion (Lane 3). The SDS–PAGE again showed that the β-Lg was resistant to pepsin in gastric digestion. Lane 1: WPI–Lac without pepsin; Lane 2: 0 min; Lane 3: 5 min; Lane 4: 30 min; and Lane 5: 60 min. Lane M represents a protein marker.

**Figure 3 foods-12-00667-f003:**
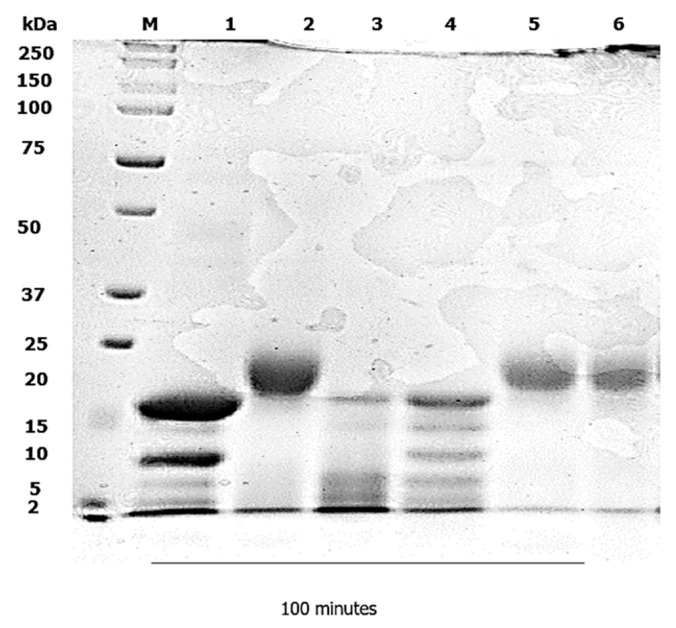
In vitro infant static duodenal digestion model of native WPI using pancreatin or without pancreatin: Lane 1 represents WPI without pancreatin; Lane 2 represents 0 min; Lane 3 represents 5 min; Lane 4 represents 30 min; Lane 5 represents 60 min; and Lane 6 represents 100 min. Lane M represents the protein marker.

**Figure 4 foods-12-00667-f004:**
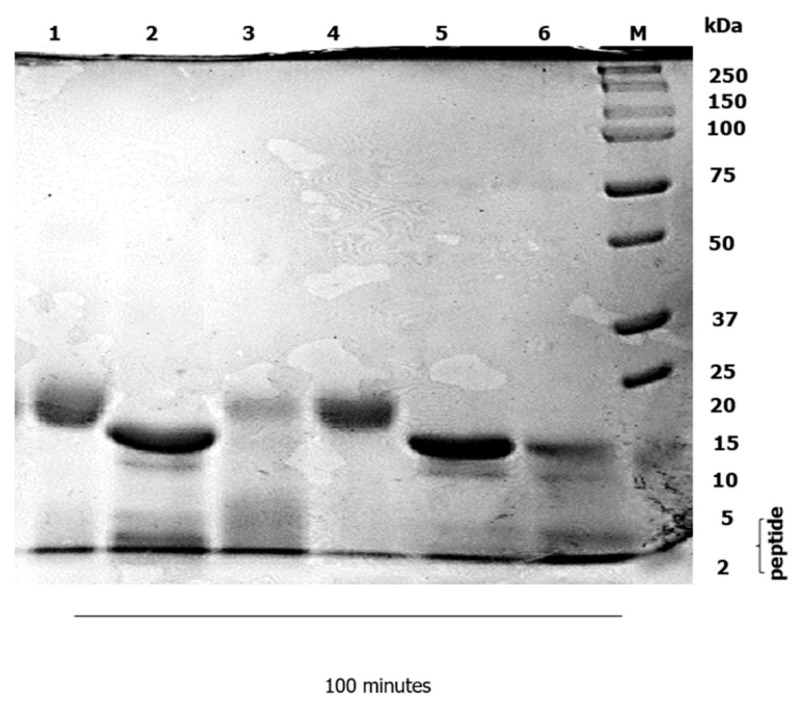
SDS–PAGE showed in vitro infant digestion of conjugated protein, using pancreatin and without pancreatin. The β-Lg was digested within 60 min (Lane 5), and after 100 min (Lane 6) there was no remaining β-Lg intact. Lane 1 represents WPI–Lac without pancreatin; Lane 2 represents 0 min; Lane 3 represents 5 min; Lane 4 represents 30 min; Lane 5 represents 60 min; and Lane 6 represents 100 min. Lane M represents the protein marker.

## Data Availability

The data are available from the corresponding author.

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
