# Peer review of "In Vitro Infant Digestion of Whey Proteins Isolate–Lactose"

_foods, 2023, doi:10.3390/foods12030667_

Round 1

Reviewer 1 Report

 In this paper, the protein digestibility of whey proteins isolate-lactose conjugates using an in-vitro infant digestion static model was introduced. Authors explored degradation of proteins in the simulated infant gastrointestinal tract to understand the digestibility of the conjugated whey proteins isolate with lactose.  It is a meaningful work to search for new infant formula with minimum impact on the allergenicity of dairy proteins and improved digestibility of protein in infants, but there are doubts in this work. In some cases, the authors provide statements regarding the allergenicity of dairy protein without supporting them or explaining. The authors need to revise the manuscript.

Comments

 Abstract. Please clarify α-Lac and β-Lg in the text.

 Lines 95 – 96. Please clarify the composition of whey protein isolate and lactose (dry matter content, protein or lactose content, ash content, etc).

 Lines 104-105. More precise description of whey protein isolate - lactose conjugation method would be helpful to understand the results. I missed the evidence that you got the conjugated protein in your experiment.

 There should be more details about the digestion process. It remains unclear from the description given by the authors how much of the sample was digested. Why was there no mouth phase designed during in vitro infant digestion?

 Line 122 -123. “The method of in-vitro duodenal digestion was carried out in accordance to the method with minor modifications “. I think there is a mistake here. The authors had in mind „in-vitro duodenal digestion “.

 Please, specify how many times the experiment has been performed.

 Lines 172 – 252. The section of the results contains only 4 figures without any description and seems to be incomplete. I think some statements from the section Discussion can be used there.

One of highlights of this paper was “to study the digestibility of the conjugated whey proteins with lactose to better understand the modified molecular structure “. More discussion of this part would be necessary.

 I would like to suggest adding more parameters for the analysis of protein degradation in the gastrointestinal tract in infants. For example, degree of protein hydrolysis.

 Lines 294-295. „Modified protein can cause the β-Lg to unfold or aggregate. These lead to an impact on protein digestion and absorption, as well as protein accessibility to the gastrointestinal immune system in the body.“ Please add a citation in this part.

  Lines 353 – 354. „The conjugation of protein with lactose enhances the susceptibility to digestive proteolysis by more accessible for the enzymatic digestive attack to the cleavage sites of the protein “. Please elaborate how do you know that?

Author Response

Prof/Dr/Sir,

The authors would like to thank the reviewer for their valuable time and comments. We have carefully considered all of the comments. Our response summarises the corresponding changes and refinements made in the revised paper except for the reviewer's suggestion to include more parameters, such as the degree of protein hydrolysis. For the time being, we did not conduct a degree of protein hydrolysis experiment in the laboratory.

Thank you.

Reviewer 2 Report

The study was conducted on the digestibility of whey protein isolates in infants and newborns. The article is appropriate when the corrections mentioned below are made.

1. Information about the maillard reaction is given in the introduction part of the article, but especially in the material and method section or in the result and discussion, details about the maillard reaction are not given.

Is the Maillard reaction formed by heat in the preparation of whey protein conjugate? Does hmf occur as a product? Hmf level of whey protein isolates should be determined by analyzing hmf. If no improvement can be made in this direction, the maillard reaction should be removed from the keywords.

2. In the materials and methods, line 138, "in 100 L (0.1 mL)" is this 100 liter or microliter?

3. line 139, "3333.3 L" same, liter or microliter. this could be checked.

4. line 169, KDa must be written in parentheses

5. line 388, "6. Patents" should be removed.

Author Response

Prof/Dr/Sir,

The authors would like to thank the reviewer for their valuable time and comments. We have carefully considered all of the comments. Our response summarises the corresponding changes and refinements made in the revised paper except for the reviewer's comment about the Hydroxymethylfurfural (hmf). For the time being, we did not analyze the hmf as the product in the Maillard reaction. We would like to point out that the Maillard reaction also produces other products based on the sample preparation and other parameters employed in the experiment conducted.

Thank you.

Round 2

Reviewer 1 Report

I have no comments for the revised version of the article.